# Functional Significance of Selective Expression of ERα and ERβ in Mammary Gland Organ Culture

**DOI:** 10.3390/ijms222313151

**Published:** 2021-12-05

**Authors:** Rajendra G. Mehta

**Affiliations:** IIT Research Institute, 10 West 35th St., Chicago, IL 60616, USA; rajumehta47@gmail.com

**Keywords:** estrogen receptor knockout (ERKO), mammary glands, epidermal growth factor (EGF), microarray estrogen responsive genes, mammary alveolar lesions (MAL)

## Abstract

Thoracic pair of mammary glands from steroid hormone-pretreated mice respond to hormones structurally and functionally in organ culture. A short exposure of glands for 24 h to 7,12 Dimethylbenz(a)anthracene (DMBA) during a 24-day culture period induced alveolar or ductal lesions. Methods: To differentiate the functional significance of ERα and ERβ, we employed estrogen receptor (ER) knockout mice. We compared the effects of DMBA on the development of preneoplastic lesions in the glands in the absence of ERα (αERKO) and ERβ (βERKO) using an MMOC protocol. Glands were also subjected to microarray analyses. We showed that estradiol can be replaced by EGF for pretreatment of mice. The carcinogen-induced lesions developed under both steroids and EGF pretreatment protocols. The glands from αERKO did not develop any lesions, whereas in βERKO mice in which ERα is intact, mammary alveolar lesions developed. Comparison of microarrays of control, αERKO and βERKO mice showed that ERα was largely responsible for proliferation and the MAP kinase pathways, whereas ERβ regulated steroid metabolism-related genes. The results indicate that ERα is essential for the development of precancerous lesions. Both subtypes, ERα and Erβ, differentially regulated gene expression in mammary glands in organ cultures.

## 1. Introduction

Mammary gland proliferation and differentiation is regulated by the coordinated action of steroid and protein hormones. Numerous reports have indicated that estrogen, progesterone and prolactin play major roles in proliferation of mammary gland into alveolar structures, whereas hydrocortisone is primarily responsible for functional differentiation of mammary glands during lactation. The mammary gland structure regresses to the ductal stage in the absence of hormones during involution in vivo by disintegrating lobulo-alveolar structures [1,2]. This entire proliferation, differentiation and involution of mammary glands can be reproduced in organ cultures of mouse mammary glands by including appropriate hormonal combinations in serum-free medium [3]. We further developed the Mouse Mammary Gland Organ Culture (MMOC) procedure to induce precancerous mammary lesions in vitro under a programmed hormonal environment in which the glands are exposed to 7,12, dimethylbenz(a)anthracene (DMBA) for a short duration of 24 h during a 24-day culture period [4]. In the presence of estrogen and progesterone in the medium, DMBA induces smaller mammary alveolar lesions, and when evaluated histopathologically the mammary glands also exhibit mammary ductal lesions [5]. Often these alveolar lesions in the absence of estrogen 17β (E) and progesterone (P) are larger and appear more aggressive than those induced with estrogen and progesterone. Hence these alveolar lesions are considered as hormone-independent lesions. These lesions have been reported to be precancerous, since transplantation of epithelial cells isolated from the lesion-bearing glands develop into mammary adenocarcinoma in syngeneic Balb/c mice [6]. 

It has been well established that steroid hormones, including estradiol, mediate their action by binding to their ligand specific nuclear receptors. Classically, the presence or absence of estrogen receptor has been used as a prime marker for distinguishing hormone-dependent breast cancers from those of hormone-independent cancers. Breast cancer biopsies exhibiting >10% cells with ER are considered ER-positive [7] and respond to Tamoxifen [8]. In 1996 another ER was cloned from the rat prostate gland and was termed as ERβ and, therefore, the classical ER was renamed as ERα [9]. Both ERα and ERβ have been cloned from a variety of tissues of human and rodent origin. Both ERα and ERβ act as ligand regulated transcription factors, and yet they are products of different genes and are located on different chromosomes; ERα is located on 6q25.1 whereas ERβ is located on 14q22-24 [10]. While both have target organ-specific independent functions, ERα and ERβ share a great portion of sequence identity within the DNA binding domain and bind estrogen-responsive elements. However, during the past 15–20 years, numerous reports have indicated that ERα and ERβ have different biological functions [11]. It has been reported that when ERα and ERβ are co-expressed, ERβ exhibits inhibitory action on ERα-mediated signaling. Thus, ERα is considered more as a proliferation regulatory receptor [12,13] whereas ERβ has a proliferation suppressive action in different tissues, including the breast [14]. This balance of ERα and ERβ expressions for carcinogenesis and cancer prevention has received considerable attention in recent years. Understanding molecular mechanisms responsible for ERα and ERβ actions are of critical importance in elucidating estrogen-mediated cellular events. The selectivity of ERα and ERβ for estrogen action can be best elucidated by using ERα or ERβ knockout mice [15]

The functional interactions between ER and EGF-R are well recognized but not well understood. There are a few reports that indicate that there exists crosstalk between these two mediators of cell proliferation [16]. In the uterus it has been reported that estradiol upregulates uterine EGF and EGF-R and that treatment with EGF mimics the growth stimulatory function of estradiol in the uterus [17,18]. Similar studies have been reported for mammary glands. Implantation of EGF pellets in ovariectomized mice results in the development of end bud in the mammary glands [19]. This concept of EGF-estrogen cross talk was proposed in MMOC by Sorof and colleagues [20]. They reported that the mammary glands can respond to growth-promoting hormones in the absence of EGF and develop expansion of mammary ducts and end bud formation. However, once the mammary glands structurally regress in the absence of hormones, similar to involution in vivo, the second round of mammary gland development requires the presence of EGF in the culture medium. Despite these isolated reports, it was not clear whether EGF can substitute estradiol for inducing estrogen responsive genes, such as progesterone receptors. 

In this study we compared systematically the morphologic similarities and dissimilarities between the hormonal responsiveness for normal and neoplastic mammary gland development amongst wild type, ERαKO and ERβKO mice. In addition, we identified clusters of genes that are either expressed in both αERKO and βERKO mice, as well as those that are selective for one of the two genotypes.

## 2. Results

### 2.1. EGF Can Replace Estradiol for the Expansion of Mammary Ducts and End Bud Formation

Mammary glands from immature mice prior to the initial estradiol surge are comprised of rudimentary mammary ducts of only 1–2 mm in length within the mammary fat pad. The treatment of both Balb/c and C57 mice with E + P daily for 6–9 days resulted in the extension of the primary mammary duct throughout the entire mammary fat pad. When we compared the length of mammary ducts in wild-type mice, in αERKO mice the ducts for all the mammary glands were rudimentary in nature and developed even to a lesser extent than the rudimentary ducts observed in wild-type Balb/c and C57BL mice. While control mice with C57 background responded to exogenous estradiol plus progesterone treatment in a manner similar to Balb/c mice, αERKO mice did not respond to estradiol as expected. However, both the wild-type and αERKO mice responded to treatment with EGF (25 ng) plus progesterone (1 mg) daily for 5 days. This resulted in extension of the ducts throughout the entire mammary fat pad. In addition, this was supported by development of end buds. These results indicate that the ductal expansion and development of end buds are dependent on ERα and not ERβ [21].

As a 25 ng EGF treatment in vivo expanded the ductal tree in the fat pad, we determined if the presence of EGF in the medium would also be sufficient to induce mammary ductal expansion. Mammary glands from the WT or αERKO mice were incubated for 10 days with IPAF +/− EGF. The glands were fixed in formalin and stained with alum carmine. Results showed no development of mammary glands in MMOC (photographs not shown). These results confirmed that the replacement of estradiol and progesterone pretreatment of glands with EGF needs to be carried out in vivo, however in vitro incubation of glands with EGF alone, without estrogen and progesterone, does not induce mammary ductal growth or development of alveolar structures.

The progesterone receptor is a well-established estradiol inducible target gene. To determine if there exists cross talk between ER and EGFR, in an earlier study we showed that EGF significantly induced PR by >4 fold (*p* < 0.01) in the mammary glands [21]. These results indicated that EGF can substitute for estradiol for the development of mammary glands or for the induction of estrogen inducible gene such as progesterone receptors. 

### 2.2. Mammary Alveolar Lesions Are Induced in βERKO Mice But Not in αERKO Mice

Earlier we reported that DMBA induces alveolar and ductal lesions in the mammary glands in MMOC. In the presence of estradiol, the lesions induced are ER positive, whereas in the absence of ER the lesions are considered as ovarian hormone independent. The alveolar lesions induced in the presence of estrogen and progesterone are usually smaller than those induced in the presence of aldosterone and hydrocortisone. As shown in Figure 1, mammary lesions induced by DMBA in mammary glands of Balb/c mice pretreated with E + P were compared with animals pretreated with EGF + P. The multiplicity of the lesions in both these groups were comparable. In E + P-pretreated mice DMBA induced 13.4 + 7.8 lesions per gland, whereas EGF + P pretreatment resulted in 12.8 + 6.8 lesions per gland. The lesions in the glands derived from mice pretreated with EGF + P were larger, denser and appeared to be more aggressive compared to the mice treated with E + P in vivo. Next, alveolar lesions induced by DMBA in wild-type C57/BL, αERKO and βERKO mice were compared. As shown in Figure 2, alveolar preneoplastic lesions were developed as expected in the wild-type C57 mice . Similar alveolar morphology was evident for βERKO mice where αER is intact. βERKO glands developed 4.7 + 3 lesions per gland. However, the mammary glands from αERKO mice, where ERβ is intact, did not develop any lesions (0 lesion/Gland) (Figure 2). These results indicated that the development of mammary lobuloalveolar differentiation and development of alveolar lesions require intact ER.ERα.

### 2.3. αER and βER Responsive Genes Are Differentially Expressed as Determined by Microarray Analyses

Pooled RNA samples of mammary tissues, after 24-day experiment to induce preneoplastic lesions in MMOC, from wild-type, βERKO and αERKO, were processed for microarray on the Code-link platform. The microarray data were analyzed for genes that were differentially expressed by >2 fold. As shown in the Venn Diagram (Figure 3A), out of 23,441 genes 2100 genes were identified as differentially regulated when αERKO and βERKO were compared. On the other hand, 1343 genes were identified when WT was compared to αERKO, and 907 genes were differentially expressed when WT was compared to βERKO-derived RNA. Overall, 723 genes were common to all the treatments. Results on the gene tree clustering of 3927 genes were subjected to a heat-map analysis. Results showed that there were significant differences between the cDNA generated from these three different types of genotype. As shown in Figure 3B,C, there was a major difference between the genes expressed in the wild type as compared to βERKO and αERKO. The genes that were overexpressed in wild type or βERKO mice were largely under-expressed in αERKO mice, whereas genes that are highly expressed in αERKO mice were only moderately expressed in wild typet and βERKO mice. 

### 2.4. Differential Expression of Genes Involved in the Steroid Receptor Signaling Pathway

A global profile of genes differentially expressed amongst these three genotypes resulted in an extensive list of genes very difficult to interpret. Therefore, we focused on the genes involved in steroid receptors and cell cycle signaling. The genes expressing differentially by greater than two-fold (up or down regulated) were separated and then divided into clusters of genes present either in both αERKO and βERKO glands and those that were selectively downregulated in either βERKO glands or in αERKO mice (Table 1). For example, genes that were down regulated by more than two-fold in αERKO, but not in WT or βERKO, were identified. A ratio was determined for αERKO/WT for the genes that were downregulated in mammary RNA of αERKO mice and normalized to WT. Similarly normalized expression for the same gene for βERKO/WT was also determined. In turn, a cumulatively normalized ratio was generated and listed as ERα and ERβ responsive genes [21].

The last column in Table 1 shows common expression in both α and β ERKO mammary RNA. This was determined as genes that were downregulated by >two-fold in both αERKO and βERKO as compared to WT.

We also identified a cluster of genes that exhibit cross regulation between ERα and ERβ. For example, as shown in the Table 2, certain genes such as TLR4, Titin, CCR3 were considered as ERα regulatory but suppressed by ERβ. These genes were downregulated in αERKO by >two-fold and simultaneously upregulated by >two-fold in βERKO. 

From the microarray results it can also be concluded that progesterone receptors were selectively identified as both ERα and ERβ responsive, whereas cytochrome C and IGFBP4 were ERβ selective. In comparison, TFF1, HNRPAB (NM_021170), cathepsin D, CCT3, HSPD1 (NM_010477) were ERα selective genes. 

### 2.5. Steroid Hormone Metabolism Pathway Analysis

In addition to selective differences between ERα and ERβ selective genes in the microarray analyses, there were several differences in a variety of clusters observed. We specifically analyzed C-21 steroid metabolism-related genes. C-21 steroid hormone metabolism regulation was compared between βERKO and wild-type mice. These results are depicted in Figure 4. 

The metabolic pathways involve formation of cholesterol from sterols, which are subsequently metabolized to pregnanolone, a crucial step for progesterone synthesis. The gene IDs represent the mediator genes regulating the selective steroid metabolism step. As shown, the differential expression of genes normalized to the expression of the wild-type sample occurs between pregnanolone and progesterone. Among the 907 genes overlapping between control and βERKO mice, C-21 steroid metabolism exhibited significant differential expression (*p* < 0.05). It was found that Cyp21A1, AKR1B1 and GE1410677 (Hydroxy δ 5-steroid dehydrogenase 3β steroid isomerase 4) were the three major differentially expressed genes between WT and βERKO mice. Since the wild type mice express both ERα and ERβ, whereas the βERKO mice do not express ERβ but expresses ERα, these genes can be considered as ERβ-responsive genes for C21 steroid metabolism. These differences may be important in understanding the differential role of Era and ERβ in steroid metabolism.

## 3. Discussion

The role of growth-promoting hormones on carcinogenesis has been studied mainly in vivo [22]. While the cell culture systems facilitate direct observations on homogenous cell types, these systems fail to represent the effects of growth-promoting factors. Mammary epithelial cells grow within the fat pad and an integrated hormonal coordination is essential for the development of mammary structures and their functional differentiation. This can be accomplished in cocultures of epithelial and stromal cells [23] or in an organ culture setting. We developed mammary gland organ culture that mimics all the physiological stages, including structural make up of the mammary gland from virgin, pregnant, lactating and post lactation mice [24]. Mammary glands in organ culture also respond to DMBA and develop precancerous mammary lesions. The MMOC allowed identification of numerous cancer chemopreventive agents such as resveratrol [25], deguelin [26], brassinin [27], sulforaphan analogs, certain vitamin D analogs ([28], and PPARγ agonists such as troglitazone, among others [29]. A summary of all the potential chemopreventive agents tested in MMOC has been published [24]. In MMOC it was also observed that the hormone-independent alveolar lesions form in the absence of estrogen and progesterone; whereas the lesions developed in the presence of estrogen and progesterone were termed as hormone dependent The differential action of ERα and ERβ has only been identified recently, and few reports can be found in the literature that focus on the differential role of ERα and ERβ in breast cancer and experimental carcinogenesis [30]. However a major shortcoming has been the unavailability of ER knockout mice for these studies. Since we have established a colony of αERKO and βERKO mice in our laboratory, we were at an advantage to evaluate the functional significance of these two ER types in mammary gland differentiation and carcinogenesis.

The characterization of ERα and Erβ-selective properties for various organs has been extensively described in several publications and reviews [31,32]. For mammary gland development, it has been established that the primary mammary ducts in αERKO mice do not extend throughout the fat pad, which is essential for the ultimate structural differentiation of mammary glands into alveolar structures and for initiating lactation. On the other hand, βERKO mice develop mammary glands similar to the wild-type, indicating that ERα is essential for mammary gland development. In the present study, we determined if EGF can induce development of mammary glands by extending mammary ducts through the fat pads. Results show that treatment with 25 ng EGF and 1 mg progesterone for 3–5 days was sufficient to induce ductal extension. The mammary glands respond to growth promoting hormones and develop end buds and alveoli. However, in comparison to the wild-type mice, the growth was not as extensive. Since glands from wild-type or αERKO were incubated with medium without estradiol, the results indicate that other αER inducible factors may be required to obtain optimal growth. 

Although ERα and ERβ have been identified and their functional significance has been established, their role in carcinogenesis is not defined, except the fact that ERα supports breast cancer growth and ERβ is cancer protective. To this end, several ERβ inducers have been identified and studied as possible modulators of carcinogenesis [33]. Using the MMOC model in this study we show that αERKO condition makes the glands unresponsive to growth promoting hormones. Since pretreatment with estradiol is a prerequisite, and estradiol action requires ERα, the lack of responsiveness to DMBA to form precancerous lesions is not due to their lack of responsiveness to the carcinogen but rather because of lack of mammary gland development. On the other hand, βERKO gland responds to DMBA and form preneoplastic lesions similar to the wild-type. These studies clearly indicate that ERα is essential in breast carcinogenesis. If ERα and ERβ have very different functions, then it is expected that these two genotypes will have altered gene regulation. To discriminate this functionality of ERα and ERβ, we analyzed microarray data generated from pooled mammary gland RNA derived from wild-type, αERKO and βERKO mice. The glands were pretreated with EGF and progesterone and incubated with DMBA for 24 h as described in the protocol. The results were compared to see which genes were differentially expressed over two-fold between αERKO and βERKO. We also compared genes that were expressed in αERKO inhibitory for ERβ action, and conversely genes expressed in βERKO inhibitory for ERα action. This discriminatory profile for ERα or β responsive genes has not been previously reported [34]. While this is a preliminary list of genes involved in the steroid signaling pathway, the genes ought to be selected and confirmed for their action. Nonetheless, it provides a clear direction for the future studies. Earlier studies examined ex-vivo cellular metabolism of estradiol via C17-oxidation, C2-hydroxulation and C16α-hydroxylation pathways in explant cultures from mouse mammary tissues [35,36] and human breast terminal duct lobular units [37]. These explant cultures represent target tissues that differ in their relative risk for developing cancer. Collectively, these data demonstrate that the target tissues can effectively metabolize steroids and suggest that the C16α-hydroxylation pathway may represent an endocrine marker for breast cancer risk.

We focused on identifying differences between the mammary RNA from the βERKO and wild type mice with regard to the C-21 steroid metabolism pathway. The role of ERβ in steroid metabolism was implicated as a possible biomarker during metastasis [38]. This is important, since progesterone synthesis is important for PR function and PR is a target gene for ER. Therefore, we focused on identifying changes occurring when cholesterol is metabolized to pregnanolone a precursor to progesterone, and hydroxy-pregnannolones. Results showed that genes regulating the formation of progesterone or androgen, and estrogen metabolism, were differentially expressed in controls compared to the βERKO genotype. The results reported on explant cultures [36,37] and presented here collectively suggest that the action of estradiol on steroid metabolism may be dependent on ERβ.

In summary, the results described in this report provide direct evidence for possible ER and EGF crosstalk where estrogen function in the absence of αER can be replaced by EGF. We show that ERα is responsible for mammary carcinogenesis, and βER has little role in the development of mammary carcinogenesis in the absence of αER. Finally, microarray analyses can provide a list of independent regulatory genes for steroid receptor signaling pathways selective for αER and ERβ. The current report provides results that allow selection of gene of interest for further confirmation and in-depth pathway analyses. The MMOC system employed here represents a novel experimental approach to identify pharmacological agents that may function as effective ERβ inducers as negative regulators of breast carcinogenesis.

## 4. Materials and Methods

### 4.1. ERKO Mice

We established colonies of C57/BL αERKO and βERKO mice in our laboratories. The breeding pairs were provided by Dr. Dennis Lubahn from the University of Missouri, Columbia, MO, USA. The animals were bred and characterized genotypically. In general, the litter size was usually smaller for the KO mice. The C57/BL mice served as controls. All mice were genotyped prior to use to confirm their ER status. The C57/BL is the only strain in which estrogen receptor knockout mice have been successfully created with genetic manipulation.

### 4.2. Effects of In Vivo Pretreatment of Animals with EGF on the Structural Differentiation of Mammary Glands in ERKO Mice

Previously, we established that in order for mammary glands to respond to hormones and/or carcinogens in organ culture, it is essential to pretreat the mice with 1µg estradiol-17β and 1 mg progesterone daily for 9 days [3]. Since mammary ducts do not respond to estrogen in αERKO mice (due to lack of ERα), without ductal expansion and endbud development the glands cannot develop fully or respond to carcinogen. Therefore, we designed studies to see if the mammary glands can respond to EGF treatment and whether EGF can replace estradiol for ductal expansion. The control and ERKO mice were treated with 25 ng EGF for 5 days and mammary glands were processed for whole mounts. The morphology for structural expansion of mammary gland ducts and end bud formation was compared with the whole mounts of mammary glands injected with 1 µg estrogen and 1 mg progesterone for 9 days 

### 4.3. Mammary Gland Organ Culture (MMOC)

The procedure for MMOC to induce preneoplastic lesions has been described in detail previously [38]. Briefly, thoracic mammary glands were aseptically dissected from 4-week-old female Blab/c, C57/BL or ERKO mice pretreated with 1 µg estrogen and 1 mg progesterone for 9 days (or 25 ng EGF and 1 mg progesterone daily for 5 days) and incubated on silk rafts in chemically defined serum free Weymouth 752/MB culture medium containing appropriate hormonal combinations. The carcinogen, 7,12 dimethylbenz(a)anthracene (DMBA 2 µg/mL or 0.078 mM) was added for 24 h on day four for the induction of precancerous lesions. Inclusion of estrogen and progesterone in the medium during the growth promoting period of the initial 10 days induced MDL (Mammary ductal lesions), whereas the presence of aldosterone and hydrocortisone during this period induce mammary alveolar lesions (MAL). The whole mounts of the glands were prepared and stained with alum carmine as previously described [29]. The experimental design is shown in the flow chart (Figure 5).

### 4.4. Microarray Analyses

Five mammary glands eachere were used from WT, αERKO and βERKO mice for the study. Mammary glands were treated with IPAF (10 days), DMBA on day 3 and 1 (days 10–24) to induce mammary alveolar lesions as described above. The glands were snap frozen in liquid nitrogen. Total RNA was extracted individually from each gland and purified using RiboPure RNA isolation (Ambion, Austin, TX, USA. Total RNA was quantitated by spectrophotometry at OD260/280 for each sample. Equal mass amounts of total RNA from each gland were pooled to obtain a pooled sample representing RNA from that group (five glands per group). Individual and pooled total RNA quality was assessed using an Agilent Bioanalyzer with the RNA6000 Nano Lab Chip (Agilent Technologies, Santa Clara, CA, USA). Biotin-labelled cRNA was prepared by linear amplification of the Poly(A)+ RNA population within the total RNA sample. Briefly, 2 µg of total RNA was reverse transcribed after priming with a DNA oligonucleotide containing the T7 RNA polymerase promoter 5′ to a d(T)24 sequence. After second-strand cDNA synthesis and purification of double-stranded cDNA, in vitro transcription was performed using T7 RNA polymerase in the presence of biotinylated UTP. Ten micrograms of purified cRNA were fragmented to uniform size and applied to CodeLink Mouse Whole Genome Bioarrays (Formerly GE Healthcare, Piscataway, NJ, USA; now supplied by Applied Microarrays, Inc, Tempe, AZ, USA) in hybridization buffer. Arrays were hybridized at 37 °C for 18 h in a shaking incubator. Arrays were washed in 0.75× TNT at 46 °C for 1 h and stained with Cy5-Streptavidin dye conjugate for 30 min. Rinsed and dried arrays were scanned with an Agilent G2565 Microarray Scanner (Agilent Technologies) at 5 µm resolution. CodeLink Expression Analysis software (Applied Microarrays, Inc, Tempe, AZ, USA) was used to process the scanned images from arrays (gridding and feature intensity extraction) and the data generated for each probe on the array were analyzed with GeneSpring GX v7.3 software (Agilent Technologies). All control genes and genes that did not pass the quality control metrics of the manufacturer were removed from further analysis. To compare individual expression values across arrays, raw intensity data from each probe were normalized to the median intensity of the array. Only genes which had values greater than background intensity in at least one condition were used for further analysis [21].

## Figures and Tables

**Figure 1 ijms-22-13151-f001:**
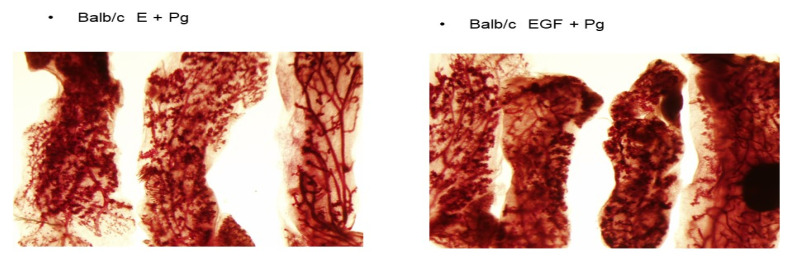
Comparison of pretreatment with E + P or with EGF + P on the induction of DMBA-induced preneoplastic lesions in balb/c mice. Mammary glands dissected from either E + P or EGF + P treated Balb/c mice were incubated with growth-promoting hormones for 10 days followed by the withdrawal of hormones to regress the mammary structures. On day three, the glands were exposed to 2 mg/mL DMBA for 24 h. The results show that the precancerous lesions were induced in both pretreatment groups. The treatment with EGF + P resulted in more dense lesions with characteristics of aggressive lesions.

**Figure 2 ijms-22-13151-f002:**
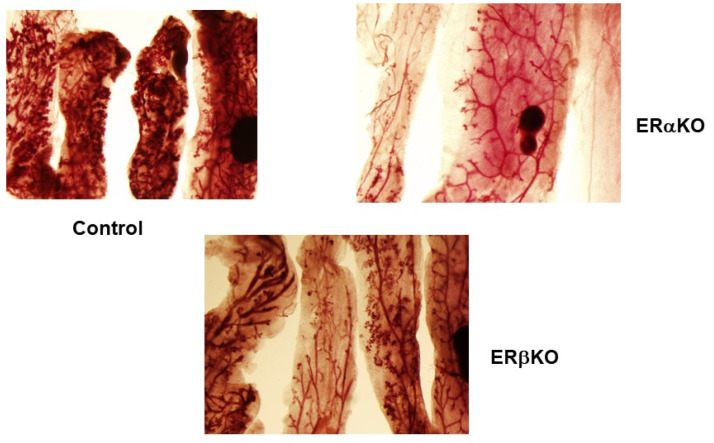
Comparison of DMBA-induced mammary lesions in C57 WT, αERKO and βERKO mice. Mice were pretreated with EGF + P for 5 days. The glands were cultured in serum-free medium with IPAF for 10 days and with I for an additional 14 days. DMBA was introduced on day 3 for 24 h. Results show that DMBA induced mammary lesions in WT and βERKO mice. In αERKO mice the mammary ducts extended throughout the fat pad; however, lesions were not formed. The results indicate that αER is essential for the induction for mammary lesions.

**Figure 3 ijms-22-13151-f003:**
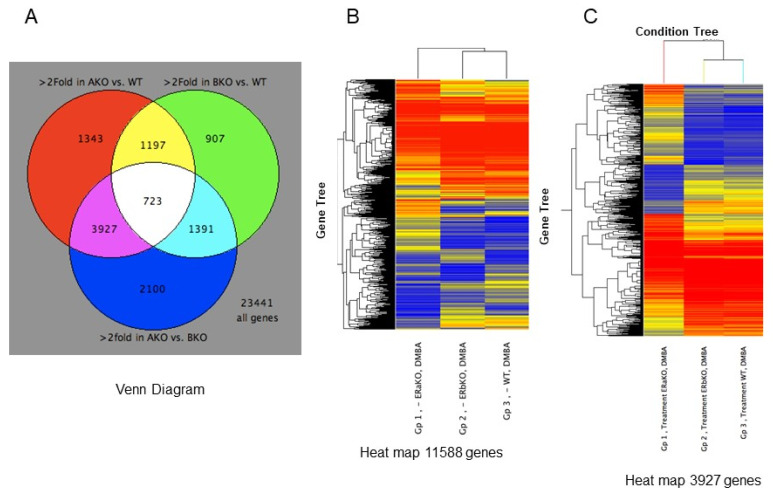
Differential gene expression in the precancerous lesions induced by carcinogen in MMOC. Mammary glands were dissected from WT, αERKO or βERKO mice pretreated with1 mg Progesterone and 25 ng EGF for 5 days. The glands were incubated with sequential combinations of hormones and carcinogen for 24 days. This treatment schedule induced preneoplastic lesions in these glands. The glands were snap frozen individually, RNA was extracted and microarray analyses were performed on pooled RNA samples as described in the Methods. (**A**) Venn diagram of genes >2-Fold in ERKO comparisons (Total 11,588 genes). The diagram indicates that 723 genes are common for mammary glands from WT, αERKO and βERKO mice. However, there are various distributions of a number of genes common between each combination. (**B**,**C**) Differentially expressed genes in WT, αERKO and βERKO mice. Genes are displayed as normalized to the median intensity of each array. Red = High expression, Yellow = Medium expression, Blue = Low expression. Results show that there is a close similarity between the expression of genes between WT and βERKO mice. However, there are major differences between the αERKO mice and the two other genotypes. These results suggest that αER may be significantly more crucial for estradiol function compared to ERβ. A part of the figure was adapted with permission from Mehta et al. [21].

**Figure 4 ijms-22-13151-f004:**
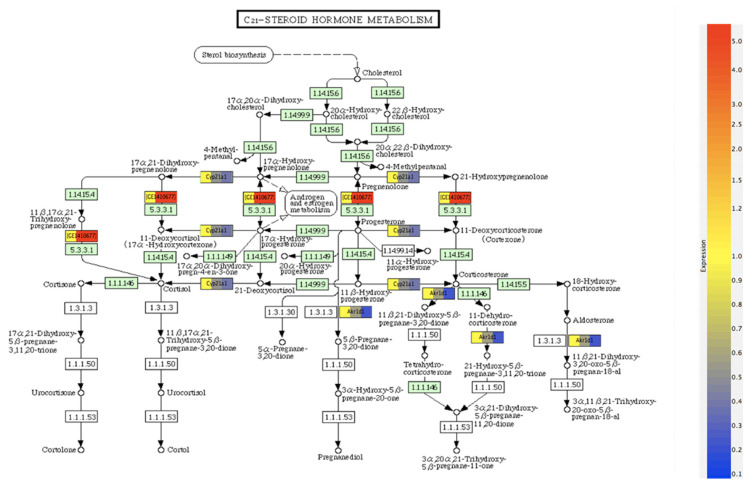
Regulation of C-21 steroid metabolism by bERKO. Differentially expressed genes (>2-Fold) in (bERKO vs. WT). Genes were normalized to the expression in the WT sample. Green boxes represent genes present on the array, not in the selected list of 2-Fold genes. White boxes represent genes not on the array. Red and blue colors represent overexpression and reduced expression of genes.

**Figure 5 ijms-22-13151-f005:**
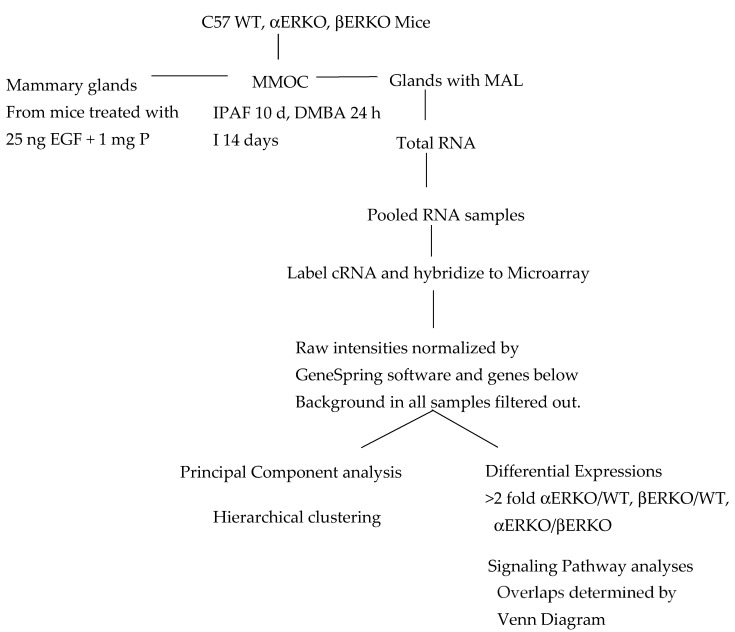
Flow chart of the experimental design.

**Table 1 ijms-22-13151-t001:** Microarray Analysis of Differential Expression of Genes Involved in the Steroid Receptor Signaling Pathway: ER Responsive Genes Regulating Cell Cycle*.

ERα Responsive Genes *	Gene ID	ERβ Responsive Genes **	Gene ID	Genes Expressed in Both ERα and ERβ KO Mice	Gene ID
BMP2 (Bone morphogenic Protein 2)	D18317	CDKi2D (p19)	NM_009878	Keratin13	NM_010662
Sash1 (Sam and SH3 domain Containing 1)	AK084957	CDKi2A (p16)	NM_009877	Protocadherin8	NM_021543
Adcyap1 (Adenyl Cyclase activating Polypeptide 1)	NM_009625	Hrasls3 (H-RAs like Suppressor 3)	NM_139269	Insulin II	NM_008387
Rbm5	AI591859	Axin2	NM_015732	Melanoma Antigen Family B5	AK006807
Cdkn2b (p15)	NM_007670	Rassf2 (Ras association domain family 2)	NM_175445	Melanoma associated Antigen	CD741902
Bcl10	NM_009740	IGFBP6	NM_008344	MMP24	NM_010808
Htatip2 (HIV-1 tat interactive protein 2)	NM_016865	Neu4 (Sialidase 4)	BB081054	Cornulin	AA794288
Cdk2ap1 (CDK2 associated protein 1)	NM_013812	Cckbr (Cholecystokinin B Receptor)	NM_007627	Aim2 (absent in Melanoma)	BC009664
PTEN (Phosphatase and tensin)	NM_008960	Itpr2	NM_019923	ADA	BB228703
Pdcd4 (Programmed cell death 4)	NM_011050	Prkcq (PKC Ө)	NM_008859	LDLR related Protein 8 (Apo E receptor)	NM_053073
Ing1 (Inhibitor of growth family 1)	NM_011919	Apoa1 (apolipoprotein A-I)	NM_009692	MMP10 (matrix metallopeptidase 10)	NM_019471
Caspase3	NM_009810	Tbxas1 (Thromboxane A Synthase 1)	NM_011539	MMP1a (matrix metallopeptidase 1a (interstitial collagenase))	NM_032006
ATM	NM_007499	Ptgis (PGI2 Synthase)	NM_008968	ADArb2 (adenosine deaminase, RNA-specific, B2)	NM_052977
Trp53 (Transformation related protein 53)	NM_011640	TRAP (Acid Phosphatase5)	NM_007388	HSPe1 (Chaperonin 10)	NM_008303
Nat6 (N acetyl transferase)	NM_019750	CD27 TNFR super family (T-cell immunity)	NM_001033126	Granzyme N	NM_153052
Tbrg1 (TGFβ regulated gene 1)	NM_025289	CD70 (CD27 Receptor) T and B cell stimulation	NM_011617	Tuberous Sclerosis 1	NM_022887
Trim13 (Tripartite mortif protein 130)	NM_023233	CCR10	NM_007721	Eph receptor A1	NM_023580
Lats1 (Large tumor Suppressor)	AF104414	Flg (Filaggrin)	XM_485270	Serine protease 3	NM_011645
VHLH	NM_009507	Pnma3 (paraneoplastic antigen MA3)	NM_153169	Protamine 3	NM_013638
BMP7	NM_007557	Tmod3 (Tropomodulin 3)	AK051327	Brevican	NM_007529
Ras HomologyB	NM_007483	Cts8 (Cathepsin 8)	NM_019541	NMDA 2A, 3A and 3B (glutamate receptors)	NM_008170 AK032394 NM_130455
		Obox5 (Oocyte specific Homeobox 5)	NM_145709	Casein β	NM_009972
		Mapk4 (mitogen-activated protein kinase 4)	NM_172632	Phosphodiestrase 4D (cAMP)	NM_011056
		VIP (vasoactive intestinal polypeptide)	NM_011702	Interferon β1 (fibroblast)	NM_010510
		Sod1 (Superoxide dismutase 1, soluble)	AI510255	Prl4a1 (Prolactin family protein)	NM_011165
		Rhox2 (reproductive homeobox 2)	NM_029203	Decay accelerating factor 2	NM_007827
		Toll-like receptor 1	NM_030682	Oxytocin	NM_011025
				Versican	XM_488510

* = Genes expressed in βERKO mice; ** = Genes expressed in αERKO mice.

**Table 2 ijms-22-13151-t002:** Clustering of αER and βER selective genes.

Genes with >2-Fold Downregulation in αERKO and >2-Fold Increase in Glands from βERKO Mice	Gene ID	Genes with >2-Fold Downregulation in βERKO and >2-Fold Increase in Glands from αERKO Mice	Gene ID
CCR3	NM_009914	IGFBP6	NM_008344
CXCL2	NM_009140	Cadherin9	BU610040
Relaxin1	AK028199	CXCR6	NM_030712
MMP8	NM_008611	TNF10	BF714828
TLR4	NM_021297	Cathepsin R	NM_020284
Transthyretin	NM_013697	Cholecystokinin B Receptor	NM_007627
Developing brain homeobox 1	NM_001005232	Albumin 1	NM_009654
Titin	BY725718	Ranbp6 (RAN binding protein 6)	NM_177721

## Data Availability

Data stored at the IIT Research Institute’s Archived folders.

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
