# Peer review of "Functional Significance of Selective Expression of ERα and ERβ in Mammary Gland Organ Culture"

_ijms, 2021, doi:10.3390/ijms222313151_

Round 1
Reviewer 1 Report
This report by Mehta., provides direct evidence for possible ER and EGF cross talk where the estrogen function in the absence of αER can be replaced by EGF. Results showed αER is responsible for mammary carcinogenesis and βER has little role in the development of mammary carcinogenesis in the absence of αER. Also, he demonstrated microarray analyses can provide a list of independent regulatory genes for steroid receptor signaling pathways selective for αER and βER. Nine miRNAs with induced expression were strongly correlated with the RCC prognosis. However, the author needs to address the following concerns.
- The abstract was not well written and can be improved.
- Besides the microarray analyses, have author had performed any gene and protein expression or immunohistochemical studies to validate the microarray results
- What proportion do authors agree on this current approach would fit with an existing standard prognosis of disease?
- The use of more ERKO organoid cultures through different mice strains in this study would have been more informative and conclusive.
- The manuscript can be revised further for grammatical and typological errors.
Author Response
Response to the reviewers’ comments
I would like to thank the reviewer for his/her constructive comments. I have made all the necessary suggested changes in the revised manuscript. Following is the response to the reviewers’ comments point by point.
Reviewer # 1
Comments and Suggestions for Authors
- The abstract was not well written and can be improved.
I have made several changes in the Abstract to improve it. Due to the word limit for the abstracts sometimes the summary does not look succinct. But I have made changes to include all the important points as suggested by the reviewer.
- Besides the microarray analyses, have author had performed any gene and protein expression or immunohistochemical studies to validate the microarray results
This is an important point. The microarray was done primarily to differentiate the gene expression related to ERa or ERb in the organ cultured mammary glands. We did not select any genes from the list to validate the results for the present study. That would be beyond the scope of the manuscript.
Previously qRT-PCR studies were carried out to see if EGF can induce progesterone receptors. The results had shown that both estrogen and EGF induced PR. These results were previously published (Mehta et al. Plos One [21]) and therefore were not included here. Secondly, we also observed induction of GPR30 a protein involved in EGF transactivation by EGF in the glands of ERa KO mice using immunohistochemistry procedures (Mehta et al. Plos One (21)). Since these results are already published they are not included in the manuscript.
- What proportion do authors agree on this current approach would fit with an existing standard prognosis of disease?
It is very difficult to answer this hypothetical and yet very interesting question. Currently there is no routine practice to determine ERa and ERb selectively in breast cancer patients. Use of ERb enhancers has been proposed as a therapeutic approach. Whether the results presented here can directly affect the prognosis may require several in depth studies and cannot be concluded from the present report
- The use of more ERKO organoid cultures through different mice strains in this study would have been more informative and conclusive.
I agree, the reviewer is correct, similar studies using different strains of ERKO mice would be more informative. However so far, the only success Dr. Korack and his group had, was in generating ERa and ERb knockout mice using C57/BL strain of mice. There is no other strain of mouse with ERKO status is available.
- The manuscript can be revised further for grammatical and typological errors
The manuscript is critically revised by correcting grammatical and typographical errors.
I hope the response and corrections will make the revised manuscript acceptable for publication. Once again, I thank the reviewer for excellent comments and suggestions.
Rajendra Mehta
Corresponding Author
Reviewer 2 Report
Study entitled „ Functional significance of selective expression of ERa and ERb in mammary gland organ culture“ is original study and provides the scientific novelty of the research. Results are comprehensively described and well discussed. Before publication, only minor comments must be implemented:
L132, correct …Figure 2 (not Figure 3),
L165, correct….Figure 3B (not 4B), also insert Figure 3C into the text,
Conclusions (last paragraph of Discussion)…regarding microarray analyses, summary is too general, please be more concrete.
I found numerous typing errors throughout the manuscript. Please carefully check!
Author Response
Reviewer No. 2.
I would like to thank the reviewer for his/her constructive comments. I have made all the necessary suggested changes in the revised manuscript. Following is the response to the reviewers’ comments point by point.
Comments and Suggestions for Authors
Study entitled „ Functional significance of selective expression of ERa and ERb in mammary gland organ culture“ is original study and provides the scientific novelty of the research. Results are comprehensively described and well discussed. Before publication, only minor comments must be implemented:
L132, correct …Figure 2 (not Figure 3),
L165, correct….Figure 3B (not 4B), also insert Figure 3C into the text,
Conclusions (last paragraph of Discussion)…regarding microarray analyses, summary is too general, please be more concrete.
I found numerous typing errors throughout the manuscript. Please carefully check!
As suggested by the reviewer, all the corrections are made, and parts of the Conclusion section is rewritten to include specific genes rather than general statements.
I hope the response and corrections will make the revised manuscript acceptable for publication. Once again, I thank the reviewers for their excellent comments and suggestions.
Rajendra Mehta
Corresponding Author